# COVID-19 Vaccine Hesitancy among English-Speaking Pregnant Women Living in Rural Western United States

**DOI:** 10.3390/vaccines11061108

**Published:** 2023-06-16

**Authors:** Elizabeth Cox, Magali Sanchez, Carly Baxter, Isabelle Crary, Emma Every, Jeff Munson, Simone Stapley, Alex Stonehill, Katherine Taylor, Willamina Widmann, Hilary Karasz, Kristina M. Adams Waldorf

**Affiliations:** 1Department of Health Systems and Population Health, School of Public Health, University of Washington, Seattle, WA 98195, USA; 2Department of Epidemiology, School of Public Health, University of Washington, Seattle, WA 98195, USA; 3School of Medicine, University of Washington, Seattle, WA 98195, USA; 4Department of Psychiatry, University of Washington, Seattle, WA 98102, USA; 5Department of Communication, University of Washington, Seattle, WA 98195, USA; 6Department of Obstetrics and Gynecology, University of Washington, Seattle, WA 98109, USA; 7Department of Global Health, University of Washington, Seattle, WA 98105, USA

**Keywords:** pregnancy, vaccine, vaccine hesitancy, COVID-19, social media, rural medicine

## Abstract

This mixed-method study investigated vaccine hesitancy among pregnant women living in rural western United States and their response to social media ads promoting COVID-19 vaccine uptake. Thirty pregnant or recently pregnant participants who live in rural zip codes in Washington, Oregon, California, and Idaho were interviewed between November 2022 and March 2023. Interviews were transcribed and coded, while the ad ratings were analyzed using linear mixed models. The study identified five main themes related to vaccine uptake, including perceived risk of COVID, sources of health information, vaccine hesitancy, and relationships with care providers. Participants rated ads most highly that used peer-based messengers and negative outcome-based content. Ads with faith-based and elder messengers were rated significantly lower than peer messengers (*p* = 0.04 and 0.001, respectively). An activation message was also rated significantly less favorably than negative outcome-based content (*p* = 0.001). Participants preferred evidence-based information and the ability to conduct their own research on vaccine safety and efficacy rather than being told to get vaccinated. Primary concerns of vaccine-hesitant respondents included the short amount of time the vaccine had been available and perceived lack of research on its safety during pregnancy. Our findings suggests that tailored messaging using peer-based messengers and negative outcome-based content can positively impact vaccine uptake among pregnant women living in rural areas of the Western United States.

## 1. Introduction

Vaccines play a critical role in protecting pregnant women and infants from adverse disease-related outcomes. Pregnant women experience greater susceptibility to many diseases and infections than non-pregnant women, leading to an increased risk of severe illness, adverse pregnancy outcomes such as preterm births or miscarriages, and death [1,2,3,4]. In fact, maternal infections are responsible for approximately 20 percent of stillbirths [5]. Newborn infants are also more vulnerable to infectious disease. The World Health Organization (WHO) estimates that 37 out of every 1000 children under the age of five die in childhood, with infections accounting for a large proportion of deaths [6]. Infants under six months of age are particularly vulnerable to infections as their immune systems are not yet fully developed [7,8]. For this reason, both the WHO and the Center for Disease Control (CDC) recommend that women receive a number of vaccines during pregnancy, including vaccines to prevent pertussis, tetanus, diphtheria, polio, influenza, and COVID-19 [9]. These vaccines provide protection for fetuses and young infants via the transfer of vaccine-specific antibodies from mother to child through the placenta or breastfeeding [10].

Despite recommendations from public health and medical leaders, vaccine hesitancy remains high among pregnant women and has been exemplified by the recent COVID-19 pandemic. COVID-19 vaccine coverage among pregnant women has lagged behind the rest of the adult population [11,12,13], even though COVID-19 is associated with an elevated risk for many perinatal adverse outcomes, including preterm birth, preeclampsia, thromboembolic disease, admission to an intensive care unit, mechanical ventilation, maternal mortality, and stillbirth [3,14,15,16,17,18,19]. Most of the research on vaccine hesitancy has focused on urban populations, which are near major United States (U.S.) academic centers. Less is known about the factors driving vaccine hesitancy in rural areas. People living in rural areas are less likely to be vaccinated for COVID-19 and expressed greater hesitancy toward getting vaccinated against COVID-19 before the vaccines were released [20]. Populations living in rural areas tend to be older, less likely to have health insurance, and more likely to have underlying medical conditions while also living further away from medical facilities, placing them at a higher risk for adverse COVID-19 outcomes [8]. Higher COVID-19 incidence and mortality rates were also present in rural areas compared to non-rural areas from the beginning of the pandemic through early 2022 [21]. In rural U.S. areas, access to Medicaid coverage for uninsured pregnant women represents another major barrier to vaccination during pregnancy. In states not offering full Medicaid coverage during pregnancy, immunization coverage for influenza and Tdap was found to be 12% and 20% lower for pregnant women living in rural areas than among their urban counterparts [22]. It is possible that for pregnant women living in rural areas, these factors compound to increase vaccine hesitancy and lower vaccine uptake rates.

The objective of this mixed methods study was to understand the factors that contribute to COVID-19 vaccine hesitancy among pregnant women in rural areas in the western U.S. Data from this study will inform public health communication campaigns targeted at vaccine-hesitant pregnant women, as well as provide recommendations for maternal public health.

## 2. Materials and Methods

### 2.1. Study Design

A mixed-methods study was chosen for the design of this research to capture factors influencing vaccine hesitancy among pregnant women, as well as their reactions to social media ads. The qualitative component consisted of direct interviews focusing on that participant’s experiences and views on vaccination while pregnant. The quantitative component reflected the participant’s self-rated reaction to several social media ads promoting vaccination.

### 2.2. Key Informant Interviews

Prior to developing the interview guide, we conducted three key informant interviews with medical professionals; they provided valuable information on their experiences of vaccine hesitancy in pregnant women that helped shape the direct interview guide (Appendix A).

### 2.3. Participants and Procedures

Our study population consisted of 30 participants living in rural Washington, Oregon, Idaho, and California, who were at least 18 years of age but younger than 40 years of age; the participants were either pregnant at the time of the interview or had given birth within the previous six months. Participants for our study were recruited via ads on Facebook and Instagram, which were targeted to rural zip codes or counties with populations of 5500 people or fewer in Washington, Idaho, Oregon, and California. In addition to targeting age and location, the ads also targeted people who had shown interest in other pregnancy- or baby-related items, (cribs, diapers, etc.). The target audience was estimated to be approximately 261,700–307,800 people and included both pregnant and non-pregnant individuals as targeting only pregnant women was not possible. The ads received 481 link clicks to open a REDCap survey, where the respondents self-verified that they met the study criteria and supplied their zip code. Our researchers verified the respondents’ eligibility via this method and contacted everyone who filled out a contact form and was eligible to schedule an interview with them. All eligible participants who responded to our outreach to schedule an interview were included. Once 30 eligible participants had been interviewed, subject recruitment was closed; these social media ads cost approximately USD 500”.

### 2.4. Measures and Instruments

Study participants completed a 14-item online survey, which included demographic questions on race, income, education level, employment status, marital status, political affiliation, religious affiliation, and COVID-19 vaccination status. Respondents then participated in individual 45–60-min guided interviews over Zoom. These interviews consisted of 10 open ended qualitative questions, which focused on individuals’ health decision making during pregnancy, trusted sources of information on COVID-19 and COVID-19 vaccines, and attitudes and experiences regarding vaccines (Appendix A). Participants were also shown four sample ads at the end of each interview and asked one closed-ended Likert scale question and one open-ended question about each ad. Upon completion of the interviews, participants received a $50 Amazon gift card, distributed to their email addresses.

### 2.5. Ad Design

We designed 19 social media ads promoting vaccination to feature one of four unique messengers (peer, doctor, elder, faith leader) and one of five content types (appeal to protect, text heavy, social proof, information on negative outcomes, or activation). An ad appealing for protection was targeted at protecting the family or the fetus from the harm of COVID-19. The text heavy ads provided a short list of persuasive reasons for becoming vaccinated, with a generally positive framing. The social proof ads conveyed the message that thousands of other pregnant women had been safely vaccinated. An ad featuring information on negative outcomes emphasized the increased risks in pregnancy for unvaccinated women. Finally, the activation ad was meant to “nudge” the participant to get the vaccine and serve as a reminder for the person who had already decided to become vaccinated but had yet to follow through. Some combinations of messenger and content type were unrealistic and, therefore, not used (i.e., elder + social proof, faith + information on negative outcomes).

Ad design was based on several marketing principles, such as selling point identity, fear appeal, and belongingness. They were designed to be original, truthful, and informative. Given the wide range of messengers and content, the images and linguistic choices were carefully selected to match the respective target audience. The ad designs were created to reflect different messengers and content types. These concepts were developed by the research team based on previous vaccine acceptance research. The photos display rural outside settings as well as other women and doctors representing the target audience themselves. The ad design utilizes a mixture of cartoon and realistic photos to further test receptiveness to visual stimulants. The faith-based messenger was aligned with the respondent’s selected faith, based on an earlier question in the REDCap survey, which triggered branching logic to show either a priest or pastor depending on the participant’s selection of faith.

Some of the ads were pre-tested in a series of iterative pilot campaigns that were promoted to urban populations of women via Facebook and Instagram during the COVID-19 pandemic (e.g., peer and doctor messengers, informational and fear-based messages). Based on viewer impressions and comments, we developed hypotheses regarding effective messengers and ad messages, which were subsequently tested in this research study. Successful ads from the pilot campaigns were then redesigned to reflect the perspective of a rural population.

In this study, the ads were presented to the participants individually via Zoom. The order of the messenger type in the ads was randomized in each ad set, while the order of the content type was not. After showing the participants each ad, the interviewer asked an open-ended question about their initial reaction to the ad. The interviewer then asked a close-ended question regarding whether the participants would be more likely to receive a COVID-19 vaccine or booster during pregnancy after seeing the ad. The participants were able to respond on a five-point Likert scale ranging from “Strongly Agree” to “Strongly Disagree”.

### 2.6. Data Analysis

Direct interviews involve prolonged engagement with and triangulation of data collection via qualitative and quantitative methods. We used a mixed-method approach to analyze qualitative and quantitative data. For the qualitative analysis, each interview was blind coded twice in Dedoose using a thematic codebook (Appendix A) by two different coders. We followed an iterative approach to developing the codebook, whereby codes were initially developed based on existing knowledge, research, and the interview guide, and then revisited and refined to capture emerging themes in the transcripts fully and accurately. The researchers kept notes regarding the coding framework and the respondents’ perspectives. Team members were encouraged to engage with the analysis as a witness to the respondents’ experience and perspective and to remain vigilant regarding their own thoughts and beliefs. Themes and subthemes were vetted by the team members. Finally, the codebook was defined before we performed an extensive analysis. We did not compute inter-rater reliability, but the coders met after the blind coding to discuss discrepancies. Disagreements in code assignment were rare and resolved by discussion.

For the quantitative analysis, we assessed the effect of messenger type and content type on ad ratings using linear mixed models (R packages “lme4” and “lmeTest”) as each participant viewed and rated four different ads on a Likert scale. We ran a separate model for each independent variable as the combination of messenger and content type was not evenly distributed and some messenger/content combinations were not represented. Messenger type and content type were set as fixed effects and participant as a random effect; no covariates were included in the model. “Peer” and “Negative Outcomes” were set as the reference categories for the messenger and content variables, respectively, as these options were rated most favorably.

## 3. Results

### 3.1. Participants

Results from this study captured data from 30 participants who participated in direct interviews (Table 1). Most participants were white, married, employed (either full-time or part-time), not religious, and held a bachelor’s degree. The majority were recently pregnant (within the last 6 months) and were vaccinated for COVID-19 at the time of the interview. A slight majority reported their political affiliation as slightly or very conservative.

### 3.2. Quantitative Analysis of Social Media Ad Reactions Promoting Vaccination

#### 3.2.1. Overview of Ads and Participants Viewing Ads

To determine respondents’ opinions on social media ads’ messengers and content, we created 19 sample social media ads, which tested four types of messengers: peer, doctor, elder, and faith leader, and five types of content: activation, social proof, text heavy, appeal to protect, and information (negative outcomes) (Figure 1). Ads were randomized by creating 10 ad sets, showing four ads each with different combinations of messenger and content types (Table 2). To evaluate participants’ reactions to ad messenger and content, we asked participants one qualitative and one quantitative question in response to each ad. First, a qualitative question was asked to determine the initial reaction to each ad, including what the respondent liked and disliked about it. The quantitative question asked respondents to rate each ad on a 1–5 Likert scale of whether they would be more or less likely to receive the COVID-19 vaccine after seeing the ad.

#### 3.2.2. Quantitative Analysis

Results from the Likert scale questions were analyzed in a mixed-effects model. First, we determined the respondents’ preferences as to the messenger in ads promoting vaccination (Figure 2A). Peer was selected as the reference category as it was the most favorably ranked messenger. Ads depicting elders and faith leaders were rated significantly less favorably than ads depicting peers (*p*-value= 0.04 and 0.001, respectively; Table 3). Ads depicting doctors were rated similarly to those of elder messengers and less favorably than peer messengers. Secondly, we investigated respondents’ preferences to the ad content (Figure 2B). Negative outcome-based ads were set as the reference category as they were the most favorably ranked content type. Activation-based content designed to nudge the participant into receiving vaccination was the only content type rated significantly less favorably than Negative outcome-based ads (*p*-value = 0.001, Table 3).

#### 3.2.3. Qualitative Reactions to Social Media Ads Promoting Vaccination

##### Participants Favored Ads Featuring Scientific Content

Participants felt positively about ads that provided facts or science-based information. Statistics provided participants with concrete evidence supporting the ads’ recommendation to receive the COVID-19 vaccine. In response to one advertisement that stated, “Hundreds of thousands of pregnant women have safely received COVID-19 vaccines and boosters to protect themselves and their babies”, a participant commented: “I think this is encouraging, when you hear hundreds of thousands pregnant women have safely received the COVID-19 [vaccine]. It’s trying to tell me if I should take it, there’s no harm in it.” Data provided participants with a context on the extent of research supporting the efficacy of the vaccine. In response to a text-heavy content ad that provided four facts on the effect of the vaccine and COVID during pregnancy, a respondent said: “I really like the four points. They’re persuasive. They’re kinda all the things that you want to hear when you’re deciding whether to get a vaccine during pregnancy and I think it would…increase my likelihood of getting vaccinated.” In response to a negative outcome-based ad with statistics on the risk of stillbirth from COVID-19, one person stated: “It is kind of intense… But it would speak to me… So reading that would have like absolutely convinced me”. These ads spoke directly to real concerns that pregnant women had and appealed to their desire to make decisions based on sound evidence.

Favorable reactions to fact or statistic-based advertisements were augmented by provision of a website address. Participants felt that this would allow them to seek more information and ascertain for themselves whether the site and facts were credible. They felt that facts and statements in the ads were most credible if there was further information to back them up. In reaction to an ad including a link to the One Vax Two Lives website, a resource providing further scientific information on the vaccine, one participant stated: “I like how there’s a website at the bottom, because I hate when they tell me that there’s research that supports something, and then, like don’t, give me anything to back it up... It would get me to read the website, and so it would probably inch me a little closer to getting the vaccination.”

This sentiment of appreciating the opportunity to research information further was shared both by participants who agreed with the information presented, and those who expressed skepticism. When reacting to a text-heavy advertisement with four facts, one reading: “The vaccine has no effect on your fertility”, one participant expressed: “So if you were like ‘what? it has no effect on my fertility? like I don’t believe that!’ like if I wanted to, I could go to that website.” The provision of resources to explore data further gave participants the autonomy to investigate on their own. The advertisements were a catalyst for individual research and directed participants to a website that provided reliable peer-reviewed information.

While in some cases the statistics provided participants encouragement regarding the safety of the vaccine, other advertisements using negative outcome-based tactics were also effective at either encouraging them to review information further on the website or conduct research of their own. In response to the advertisement stating “COVID increases your risk of death in pregnancy by 22× and stillbirth by 4×”, one participant commented “It’s good messaging, but it’s kind of fear based like it would kind of scare me more than encourage me, so suppose I agree that it would encourage me to get vaxxed.” These negative outcome-based advertisements integrated inclusion of statistical information with appeal to the emotional side of receiving, or not receiving, the vaccine, which proved to be an effective strategy to encourage engagement with the website and additional research for many participants. While participants expressed that the emotional appeal of the negative outcome-based advertisements would likely be effective at encouraging them to get the vaccine, their reactions to these advertisements were at times still negative. Some participants perceived these ads as fear mongering, which they strongly disagreed with as an appropriate form of advertising.

Participants responded very positively to an appeal to positive emotional reactions, such as a desire to protect their baby or family members. One participant said: “I like the message behind it, and then they look very happy. So that makes you want to smile too.” Participants also appreciated ads that they felt addressed or dispelled common myths surrounding the COVID-19 vaccine and pregnancy. Ads with text-heavy content included informative statements such as, “Pregnant people are more likely to get extremely sick from COVID-19” and “Antibodies that your body makes to the vaccine will protect your newborn”. One participant pointed out that varied facts were helpful and noted, “I like that it focuses on the pregnant person as the first point and protecting the baby as the second point.”

##### Participants Reacted Negatively toward Ads Featuring Elderly and Faith-Based Messengers and Activation-Based Content

Participants reacted most negatively toward ads that they interpreted as telling them to get the vaccine without providing facts or compelling evidence. This was true regardless of whether the messenger was a trusted elder or community physician. One participant explained, “I do not like language that says medical research is clear without proving what that research is. That feels very scammy.” To be more drawn to the ad, participants report that evidence would be needed to support the validity of the ad: “… I would find the message stronger if I saw like specific statistics on it.” Ultimately, it was the lack of evidence in the ads that led to activation ads being the only content type that was rated significantly lower than negative outcome-based content ads overall.

Most participants reacted negatively toward the faith-based messengers in the ads. This was true for both non-religious and participants who identified as having a strong relationship with faith. For many participants who identify as religious, the faith-based ads were perceived to be negative because they did not feel that their faith played a role in their medical decision making. One participant noted: “[Vaccination is] still a personal decision that you have to make on your own.” Even symbols representing a faith-based institution were a turn-off: “I think that the like cutesy like the colors are very you know. That would reel me in, but the cross would be like a straight no.” The consensus was clear that faith-based advertisements did not elicit positive reactions from participants and instead repelled them from engaging with the ad at all.

### 3.3. Qualitative Analysis of Interview Themes

#### 3.3.1. Themes Emerging from the Qualitative Analysis

Five key themes were identified in interviews, with 10 sub-themes (Table 4, Appendix A). Themes identified included vaccine uptake, self-perceived risk of COVID, sources of information on health decision-making, vaccine hesitancy, and relationship with care provider.

#### 3.3.2. Facilitators and Barriers to Vaccine Uptake

The most common facilitators to vaccine uptake that participants mentioned were wanting to keep both themselves and their fetus safe from COVID-19 and wanting to transfer vaccine antibodies to their child via the placenta or breastfeeding (Appendix A). Other respondents were motivated to get vaccinated after receiving recommendations from their care provider or having conversations with their care provider about the importance of receiving the vaccine during pregnancy. For one respondent who was initially strongly vaccine hesitant, the decision to receive the vaccine “goes back to that one-on-one conversation with our family doctor, and the research that they found with the studying of the placenta… Had we not had that conversation, and she didn’t send me that information… I probably would not have [gotten vaccinated].” It was clear that for many participants, recommendations from trusted care providers helped facilitate their decision to vaccinate during pregnancy.

Other respondents, who self-described as vaccine hesitant or vaccine neutral, stated that COVID regulations and work and travel requirements were key facilitators to their vaccine uptake. Because these participants likely would not have chosen to be vaccinated otherwise, these requirements were important facilitators to their vaccine uptake.

The most cited barrier to vaccine uptake for respondents was the perceived short amount of time that the COVID-19 vaccine had been in development and available to the public for. Respondents who distrusted the vaccine for this reason shared sentiments such as: “I had a conversation with my primary doctor about it shortly after the second time that I got COVID, and I just still wasn’t comfortable with the kind of lack of research out there… it’s just not been around long to feel like I can make an informed decision to take it”. Other respondents brought up a perceived lack of clinical trials and lack of research on long term studies or long-term effects. Some of these respondents felt that they would be willing to give the COVID-19 vaccine a chance after more research had been done. For many of these respondents, it was important for them to feel like they were doing their own research and making a well-informed decision on whether to receive the vaccine and were not convinced by the current research and evidence available.

Another common barrier to vaccine uptake among respondents was related to self-perceived low risk of COVID-19. This was expressed through common sentiments such as: “COVID didn’t seem like something that would hurt me. I don’t get the flu shot. COVID feels like the flu to me. So why would I get the COVID shot?”, as well as participants’ belief that they were already sufficiently protected from COVID without the vaccines, and able to protect their fetuses or infants as well. The prospect of needing to receive yearly COVID boosters, similar to a yearly influenza shot, was also a common barrier to vaccine uptake. Some participants felt that boosters were unlikely to protect them against COVID. Lastly, some respondents were deterred from receiving the vaccine due to their perceived politicization of it. These participants felt that recommendations or pressure to receive the vaccine were political in nature and weakened their trust in potential health benefits.

#### 3.3.3. Self-perceived Risk of COVID

Most respondents felt a greater self-perceived risk of COVID-19 during their pregnancy, compared to before or after it (Appendix A). The most common reason expressed was that respondents were worried about being more susceptible to COVID-19 themselves during pregnancy, as well as the effects that the illness may have on their fetus: “Early this year, when I wasn’t pregnant… I was a little bit carefree like, ‘Okay, bring it on. I can still try to take it’. Well, now that I’m pregnant… I don’t want anything to happen to my little boy.” Many understood that pregnant women can become more severely sick and were worried during their pregnancy about putting either themselves or their fetus at risk. Many other respondents who were unvaccinated or vaccine hesitant did not experience a change in self-perceived risk during their pregnancy compared to before or after it. One respondent said: “By the time I was pregnant, [COVID] had already been around for close to 2 years, and everything was relaxing about it... you worry about getting the flu or the cold or whatever, but nothing too serious.” Respondents who had had mild cases of COVID prior to their pregnancy were also less likely to have a high self-perceived risk of COVID during pregnancy and felt that even if they did get COVID while pregnant, it was unlikely to be severe.

#### 3.3.4. Sources of Information on Health Decision-Making

When talking about distrusted sources of information, most respondents mentioned distrusting the internet, social media, or personal opinions (Appendix A). This was commonly expressed through sentiments such as: “I use social media all the time otherwise, but for my COVID information I avoided it like the plague”. Other respondents mentioned not trusting sources that had perceived low levels of evidence, such as Facebook, and relied more on anecdotes rather than research. Overall, trust in social media was low across the board. Many others mentioned distrusting traditional media or sources that they felt leaned too strongly politically toward the right or the left. This was also true for respondents regardless of vaccination status or attitudes regarding vaccine hesitancy. One respondent said: “Illness should not be politically charged or involved, and so, therefore either right winged or left winged”. A few respondents did not trust sources within their rural communities. One respondent said: “I didn’t ask [the doctor’s] office staff questions…They are vaccinated because they have to be, but I felt that it was fairly unlikely that they were going to provide very much evidence-based information”. These participants did not feel that their local news or community leaders were equipped to provide them with up-to-date or evidence-based information.

Compared to distrusted sources, most respondents trusted their care providers, local and national public health departments, peer-reviewed studies, websites that ended in “.org” or “.edu”, or sites such as WebMD. One said: “I already have, of course, my own opinions, and have done some of my own research. But I look to [my doctor] for confirmation that I’m understanding what’s happening”. Participants looked to their care providers to help confirm their decisions and trusted that their care providers were aware of all up to date research and evidence. For some participants, self-reported trust in their care providers was not enough to overcome COVID-19 vaccine hesitancy or influence their vaccine uptake. When one participant’s doctor brought up the COVID-19 vaccine for her baby, her response was: “Don’t come at me, bro. When I have this baby, I’m not pushing this vaccine on a newborn. I won’t do it.” This speaks to the role that participants felt that their care providers played in influencing their health decisions, in that care providers could try to help inform decisions, but ultimately were not responsible for deciding what the patient should choose to do. Respondents also largely trusted sources that they felt were presenting them with evidence-based, unbiased, and up to date information that would help them make their own informed decisions. When asked what influenced their health-based decision making, most participants mentioned using their own research to inform their decisions.

#### 3.3.5. Vaccine Hesitancy

Vaccine hesitancy was most attributed to worries that the COVID vaccine had not been sufficiently researched before being released to the public (Appendix A). This was seen in statements such as: “I know there’s lots of scientific information out there, but at the same time it’s, it’s something that’s still very new”. Many of these participants were hesitant of the COVID-19 vaccine while willing to take other routine vaccines recommended to them by doctors. Some participants attributed this hesitation to previous medical advice for women that has caused harm. One participant said: “‘I’ll take the pertussis vaccine that’s been being given for how many decades when you’re pregnant. [But] there have literally been documented times in history when you’re giving pregnant women things…that you think are safe, and then decades down the road you’re like, ‘Oh, well… we shouldn’t have done that. That was a bad idea’.” This again speaks to participants’ willingness to trust routine vaccines for themselves and their children that had been out for longer periods of time, compared to the perceived short period of time that the COVID vaccine had been in development and out for.

Respondents who were vaccine hesitant were also more likely to not receive other routine vaccines, such as the seasonal influenza vaccine, or to have had prior negative experiences with vaccines. One respondent said: “We don’t really get [the seasonal influenza vaccine] because it seems like a lot of times it ends up being the wrong strain...you get the flu plenty often, and it never becomes too big of a deal”. Another respondent said: “I’ve never really gotten vaccinated for anything since I almost died as a baby getting vaccinated. I’m more afraid of the vaccine than I am of the sickness”. Others felt conflicted about making decisions surrounding vaccines for themselves while pregnant due to any potential impact on the fetus, or on making decisions for their children. This speaks to the unique decision-making factors that a pregnant person or parent must consider when making health decisions.

Trust in the COVID-19 vaccines, on the other hand, was felt by participants who had a history of receiving all their vaccines and who trusted the vaccine process. Respondents frequently compared the necessity of the COVID-19 vaccine to the flu vaccine and said things such as: “I was very convinced by what I had seen regarding its effectiveness and preventing serious illness. [The COVID-19 vaccine] was something that I didn’t really think about, any more than I thought about you know, getting my flu shot every year.” Compared to vaccine-hesitant participants who perceived the COVID-19 vaccine as being created in a short amount of time, participants who were vaccine-positive perceived the vaccines to be the product of decades of scientific work and research. One participant said: “These are just remarkable vaccines that, you know, are the product of a decades for the work and scientific research.” Respondents who were vaccine positive were also more likely to express that they would be giving their children the COVID-19 vaccination, as well as all other routine vaccines.

#### 3.3.6. Relationship with Care Providers Influences Vaccine Acceptance

The primary factors which contributed to respondents’ having negative relationships with their care providers were respondents feeling that their care providers were rushed and unresponsive or that their care providers disregarded or belittled their concerns (Appendix A). For many respondents, living in a rural community meant that provider options were limited, and for some that meant being stuck with a care provider with whom they did not have an established or trusting relationship with. For respondents who had positive relationships with care providers, contributing factors included feeling listened to and understood by care providers, having an established or personable relationship with their care providers, having their autonomy and decision-making respected by their care providers, and having care providers who were extremely responsive. Respondents valued having a care provider who took the time to listen to their concerns and remember things about them. One respondent said: “I felt like because I knew that he’s my family’s doctor, I…was able to… trust his opinion, and I didn’t feel like I needed to second guess him”. It was also extremely important to respondents that they felt that they had autonomy to make their own health decisions and that their care providers respected their right to do so. Many respondents felt confident in their own health decision making and appreciated it when their care providers shared information to help inform those decisions, without being too pushy.

## 4. Discussion

### 4.1. Summary of Study Findings

Our study is one of the first to investigate social media ad reactions to vaccine uptake in pregnancy, with a focus on rural English-speaking populations in the Western U.S. Our main study findings were that major contributors to COVID-19 vaccine hesitancy among this population were the beliefs that the vaccine had not been sufficiently researched, fears about potential side effects for the mother or fetus, and a low self-perceived risk of COVID-19. On the other hand, respondents clearly valued making health-based decisions using scientifically-sound evidence and research and considered their care providers to be trusted sources of information in making or informing those decisions. Respondents were more likely to trust and value care providers who listened to their needs, affirmed their autonomy, and took time to answer their questions. It is possible that this reflects the information-seeking phase of life, which characterizes pregnancy. Our respondents not only valued receiving information from doctors who addressed their questions and concerns but also receiving information in a way that affirmed the research they had already performed and empowered them to make the final decision for their bodies and health. One respondent said of her trusted family doctor, “I… had a baseline level of trust with him already, and was able to...trust his opinion, and I didn’t feel like I needed to second guess him”, while another respondent said, “[My doctor and I have] got a good relationship, and she knows my health history… So… I feel comfortable asking her the bigger decisions.” These participants indicated that they would be more willing or comfortable to go to their care providers for health advice. Additionally, vaccine-hesitant women who had positive and trusting relationships with their care providers might be more open to receiving vaccine recommendations.

In response to the social media test ads, respondents most favorably ranked ads that were based on negative outcomes or came from a peer messenger. Respondents significantly less favorably ranked ads that were activation-based in content or came from elder or faith messengers. These findings were supported by qualitative evidence, as the negative outcome-based ads provided facts and resources that participants could verify themselves or use to inform their decision, whereas the activation-based ads merely consisted of a messenger telling the viewer to get vaccinated. This was perceived as pushing a decision on the viewer, rather than supporting them in making their own informed decision. Similarly, qualitative findings support respondents’ significantly less favorable ratings of elder or faith messengers. Respondents again largely felt that trusted sources of information were people or groups qualified to be so, such as public health agencies, care providers, or family members who worked in related fields. Respondents tended to dislike messengers in ads who they did not see as having these qualifications, such as grandparents or leaders of faith. This corresponds with our findings from the qualitative interviews, where faith and religion were rarely brought up by respondents and were not mentioned as trusted sources of information on COVID-19 or as influential sources on health decision-making.

### 4.2. Study Findings in the Context of Literature

Our study findings are in line with other studies and research on vaccine hesitancy, vaccine hesitancy in pregnancy, and social media public health campaigns. Common sources of vaccine hesitancy have been reported include concerns about vaccine safety and efficacy, misinformation, mistrust, lack of knowledge, and perceived risk of COVID-19 [11,12,23,24,25,26,27]. In a 2020 cross-sectional survey of pregnant people in Utah, Alabama, and New York, only 41% the participants indicated that they would consent to COVID-19 vaccination [13]. These findings correspond with those of our study, indicating that concerns about the safety and efficacy of COVID-19 vaccine are high among those who are still unvaccinated.

Other studies and research have explored factors facilitating vaccine uptake among pregnant people. These have commonly been found to include recommendations from care providers, social proof, trust in other vaccines, and availability of data [13,25,28,29,30]. In one study, 36 percent of pregnant participants said that a physician recommendation would reduce vaccine hesitancy [31]. In a different study on how mothers’ perceptions of vaccines change over time, those who experienced an increased confidence in vaccines did so in part due to positive relationships with care providers and conversations with them on the safety and importance of vaccines. In our own study we saw the influential role that care providers play in health decision making, and the increased trust that participants have in care providers with whom they have positive relationships.

There are many studies on vaccine uptake and hesitancy in rural areas in the Global South (i.e., Kenya and Nepal), but very few studies in rural U.S. In a study of pregnant individuals in East Tennessee, where many participants might have lived in rural areas, partially or fully vaccinated patients were more likely to obtain information about COVID-19 from their obstetric providers and place their trust in them; belief in misinformation was also high in the unvaccinated group [32]. Similarly, misconceptions about COVID-19 vaccine recommendations among pregnant women and belief in misinformation were prevalent in remote Alaskan communities [33]. Neither of these studies investigated participant reactions to social media ads or other educational materials as part of public health communication campaigns.

Our study is unique in combining qualitative interviews on vaccine hesitancy with reactions to social media public health campaign ads, allowing us to contextualize participants’ reactions to social media ads with their own unique experiences and perspectives on vaccines and pregnancy.

### 4.3. Implications for Clinical Care or Public Health Campaigns

The results of our study have practical implications for both the clinical care and public health sectors. Our results support that care providers should focus on addressing the long history of mRNA vaccine development and safety, benefits, and lack of side effects on mother or fetus, and the current risk of contracting COVID-19 infection through evidence-based facts and statistics.

Our findings also provide interesting implications for future public health campaigns. Ads with a simple, dramatic statistic (such as the increased risk of death during pregnancy in Ad 4) outperformed the more complex or creative approaches. This reflects our hypothesis that pregnant people are in a more cautious, thoughtful decision-making state than the public when it comes to decisions such as getting vaccinated and, thus, are less susceptible to ads that nudge creatively or apply social pressure. While the public has had access to an overwhelming amount of information regarding the safety and efficacy of COVID vaccines, information specific to pregnant people is naturally more specialized and rarer, so presenting that information in ad campaigns is more impactful.

While the participants acknowledged that the negative outcome content-based ads could be perceived as a fear-based tactic, they preferred the inclusion of statistics endorsing vaccination that they could verify on their own. Fear-based messaging was avoided in public health communication campaigns until the late 1970s and 1980s, when ads were run to inform the public of the hazards of cigarette smoking and the risks of acquiring HIV/AIDS [34]. While some people saw these tactics as stigmatizing certain behaviors or communities of people, others believed that these types of campaigns provided both emotional and rational appeals to promote healthier behaviors. Our findings provide support for the idea that public health campaigns can include facts focusing on negative outcomes to encourage further research, discernment, or curiosity on the part of targeted viewers, and these campaigns can be effective in doing so without stigmatizing or turning away viewers.

These results imply that public health campaigns should focus on creating content based on easy-to-read numbers and statistics, along with dissemination of updated facts regarding vaccination during pregnancy, from a peer or trusted messenger. Social media platforms allow ad campaigns to be more responsive, diverse, and dynamic compared to past eras dominated by TV, radio, newspapers, and billboard advertising. Public health officials should take advantage of this flexibility, which allows the tailoring of ads to different demographic audiences, drawing on the very latest research and directives to update ad messages frequently. Several different messages can also be delivered that overlap and reinforce one another.

### 4.4. Strengths and Limitations

A study strength was the mixed methods design, which allowed us to use qualitative data from interviews to understand and explain quantitative results regarding social media ad ratings. Secondly, we tested the effect of ad messenger and ad content, which increased study rigor. Although most of our participants were vaccinated, they came from a highly vaccine resistant demographic, who were pregnant, politically conservative, and living in rural areas of the Western U.S. There are a few limitations to our study that we would like to acknowledge. The first is that our findings from this study are not generalizable to pregnant women in urban areas or outside of Western U.S. The majority of our participants were White and lived in rural areas. Our study might be subject to various biases due to the self-reported nature of demographic data, such social desirability bias. For example, when discussing trusted and distrusted sources of information, the participants might have been hesitant to acknowledge the true extent to which they received information from social media or were influenced by information on social media. The content type and messenger combinations of the social media ads were not equally represented, meaning that the respondents were more likely to see some ad types than others. Finally, there are disadvantages associated with thematic qualitative analysis, which are mitigated by the inclusion of quantitative results. Although a simple thematic analysis can yield inconsistent results, applying a hypothesis and combining the analysis with quantitative methods build a more robust research framework.

## 5. Conclusions

This study is the first to directly address reactions to public health social media ads in rural pregnant women, a highly vaccine hesitant population. We found that pregnant subjects preferred to make evidence-based decisions regarding their health and vaccinations during pregnancy. Participants with self-reported vaccine hesitancy were concerned about the perceived short amount of time that the COVID-19 vaccine was in development and on the market. They perceived that there was a lack of research regarding the safety of the vaccine during pregnancy. The respondents preferred social media ads which provided them with facts or evidence that they could verify for themselves, as opposed to ads which came from the perspective of someone not qualified to give them medical advice, such as an elder or leader of faith, or ads which simply encouraged them to get the vaccine. These findings are critical for building future public health campaigns to increase rates of vaccination, particularly among hesitant pregnant women. There is a paucity of research on how public health communication campaigns connect with vulnerable populations with poor vaccine uptake. The digital nature of public health communication requires collaborations between public health researchers and communication and marketing experts to find new ways to connect with the public. Pregnant individuals represent an extremely vaccine-hesitant group and are, therefore, an interesting population to test different communication strategies. The findings were interesting in that simple information-based ads with statistics and fear-based ads were rated as more effective than reminders (activation-based content) or messages from an elder or a faith leader. Future research needs to address reactions to social media ads promoting vaccination and featuring different messengers and messages among specific demographic groups to determine if these findings are universal or more specific to U.S. rural pregnant populations. Additionally, more research needs to be conducted to investigate vaccine hesitancy within non-English-speaking populations, as well as groups that are historically marginalized and harmed by medicine, as they are likely to have their own unique set of factors influencing vaccine hesitancy and uptake.

## Figures and Tables

**Figure 1 vaccines-11-01108-f001:**
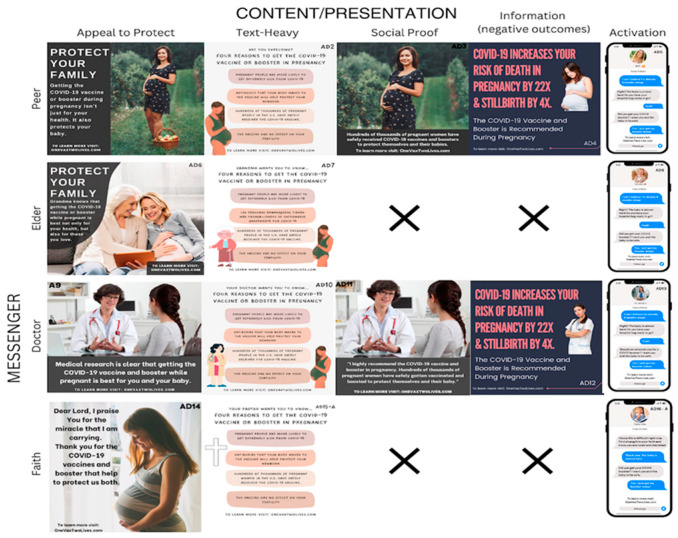
Social Media Ads Displaying Combinations of Messengers and Content Themes. This figure depicts the social ads shown to participants categorized by messenger and content type. Within a specific category, we attempted to keep the messenger or the content similar across ads to enable comparisons within a group. In some cases, the combination of messenger and content type (i.e., faith-based messenger and negative outcome-based ad) was not realistic, so these ads were not created.

**Figure 2 vaccines-11-01108-f002:**
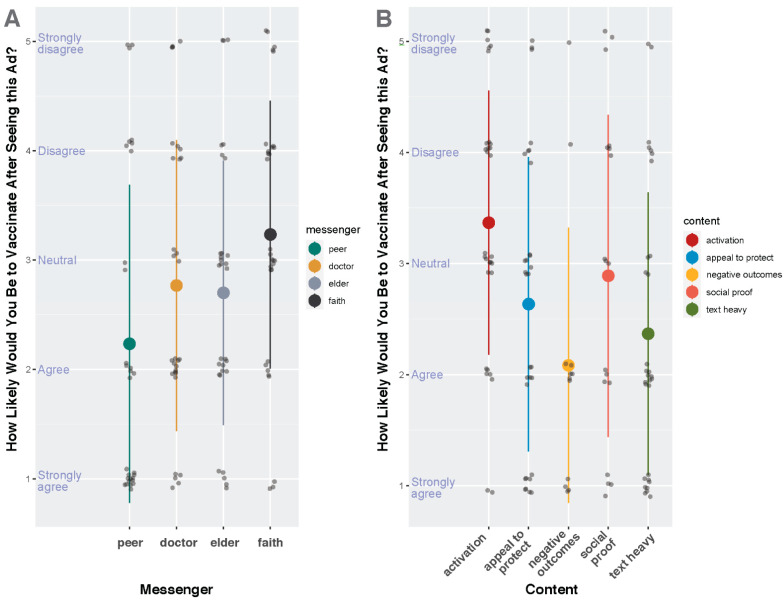
Subject Reported Likelihood of Becoming Vaccinated or Boosted After Seeing an Ad Featuring Specific Messengers and Content. This figure depicts the self-rated likelihood that a participant would receive a COVID-19 vaccine after seeing an ad showing a particular messenger (**A**) or a particular content type (**B**). The “fear” category indicated negative outcomes-based content. The Likert scale was constructed so that a rating of one indicated strong agreement that they would become vaccinated and a five rating that they strongly disagreed with becoming vaccinated. The large dot indicates the mean in each category.

**Table 1 vaccines-11-01108-t001:** Demographic Characteristics of Participants.

Characteristic	Categories	N (%)
Race and Ethnicity *	White Black of African American Hispanic/Latino American Indian/Alaskan Native	25 (83.3) 4 (13.3) 3 (10) 1 (3.3)
Pregnancy Status	Currently Pregnant Pregnant within Last Six Months	10 (33.3) 20 (66.7)
Number of Children	0 1 2–4 >4 Prefer Not to Say	4 (13.3) 14 (46.7) 9 (30) 2 (6.7) 1 (3.3)
Marital Status	Married Not Married, Living with Partner	29 (96.7) 1 (3.3)
Level of Education	Some High School High School Bachelor’s Degree Master’s Degree Trade School Prefer not to Say	1 (3.3) 8 (26.7) 12 (40) 7 (23.3) 1 (3.3) 1 (3.3)
Employment Status	Employed Full-Time Employed Part-Time Seeking Opportunities Other	10 (33.3) 8 (26.7) 4 (13.3) 8 (26.7)
Annual Household Income	<25,000 25,000–50,000 50,000–100,000 100,000–200,000	6 (20) 7 (23.3) 12 (40) 5 (16.7)
Religion	Not Religious Christian (Protestant) Christian (Catholic) Christian (Any other denomination) Other	13 (43.3) 4 (13.3) 3 (10) 8 (26.7) 2 (6.7)
Political Affiliation	Very Liberal Slightly Liberal Slightly Conservative Very Conservative Prefer not to Say	6 (20) 6 (20) 10 (33.3) 6 (20) 2 (6.7)
Vaccination Status	Received a COVID-19 Vaccine Not Vaccinated	21 (70) 9 (30)
Type of COVID-19 Vaccine Received	Type of COVID-19 Vaccine Received (N = 21) ** Johnson & Johnson Moderna Pfizer	3 (14.3) 12 (57.1) 10 (47.6)
Number of COVID-19 Boosters Received	1st Booster 2nd Booster 3rd Booster	12 (40) 6 (20) 2 (6.7)

* Respondents were able to select more than one race or ethnicity; ** Respondents were able to select more than one type of vaccine received.

**Table 2 vaccines-11-01108-t002:** Number of Subjects Viewing Specific Combinations of Ad Messenger and Ad Content.

Messenger	Ad Number	Content	Number of Views
Peer	1	Appeal to Protect	5
	2	Text-Heavy	5
	3	Social Proof	10
	4	Information (Negative outcomes)	5
	5	Activation	5
Elder	6	Appeal to Protect	13
	7	Text-Heavy	6
	8	Activation	9
Doctor	9	Appeal to Protect	7
	10	Text-Heavy	6
	11	Social Proof	9
	12	Information (Negative outcomes)	6
	13	Activation	2
Faith	14	Appeal to Protect	4
	15	Text-Heavy	11
	16	Activation	14

**Table 3 vaccines-11-01108-t003:** Effect of Messenger Type and Content Type on Ad Ratings.

Messenger	How Likely				Content	How Likely			
Predictors	Estimates	SE	t	*p*	Predictors	Estimates	SE	t	*p*
(Intercept) ^a^	2.23	0.2	9.4	<0.001	(Intercept) ^b^	2.04	0.4	5.8	<0.001
Doctor	0.53	0.3	1.9	0.063	Activation	1.32	0.4	3.5	0.001
Elder	0.47	0.3	1.6	0.103	Social Proof	0.87	0.4	2	0.047
Faith	1.00	0.3	3.5	0.001	Text Heavy	0.32	0.4	0.9	0.395
					Appeal to Protect	0.59	0.4	1.6	0.123
**Random Effects**									
σ^2^ (residual variance)	1.21				σ^2^ (residual variance)	1.15			
τ_00_ (random intercept variance)	0.50				τ_00_ (random intercept variance)	0.52			
N	30				N	30			
Observations	120				Observations	120			
Marginal R^2^	0.069				Marginal R^2^	0.103			
Conditional R^2^	0.340				Conditional R^2^	0.382			

^a^ Peer as reference category; ^b^ Negative Outcomes as reference category.

**Table 4 vaccines-11-01108-t004:** Key themes and sub-themes from direct interviews.

Themes	Sub-Themes
Vaccine Uptake	Facilitators Barriers
Self-perceived Risk of COVID-19 During Pregnancy	Greater Perceived Risk No Perceived Change in Risk
Information Source for Health Decision-Making	Trusted Sources Distrusted Sources
Vaccine Hesitancy	Vaccine Hesitant Vaccine Positive
Relationship with Care Provider	Positive Relationship Negative Relationship

## Data Availability

Data are available upon request.

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
