# Peer review of "COVID-19 Vaccine Hesitancy among English-Speaking Pregnant Women Living in Rural Western United States"

_vaccines, 2023, doi:10.3390/vaccines11061108_

Round 1

Reviewer 1 Report

Abstract:

Overall, the abstract provide a clear overview of the study's goals and methods. However, there are a few points that could be improved:

The abstract could benefit from a more detailed description of the study's findings. While the five main themes related to vaccine uptake are mentioned, it would be helpful to include specific examples of what these themes entail and how they relate to vaccine hesitancy among pregnant women in rural areas.

It would also be useful to know how significant the difference was in ad ratings between preferred and less favored messenger/content types. This would provide a better understanding of the impact that tailored messaging can have on vaccine uptake.

Introduction:

Overall, the introduction provide a good foundation for the study, but more information could be provided to improve the reader's understanding of the study's goals, methods, and findings.

While the introduction provides a good overview of why vaccines are important for pregnant women and infants, it could benefit from more specific information on the risks associated with COVID-19 during pregnancy. For example, it could mention that pregnant women with COVID-19 are at increased risk of preterm birth, preeclampsia, and stillbirth, and that the risk of severe illness and death is higher in pregnant women compared to non-pregnant women.

It would also be useful to know how the study's findings contribute to the existing literature on vaccine hesitancy among pregnant women, particularly in rural areas. Are there any notable differences between this study's findings and previous research on the topic? What implications do these findings have for public health communication campaigns targeting vaccine-hesitant pregnant women in rural areas?

Methodology:

Overall, the study design and methodology seem well-planned and appropriate for the research questions. However, there are a few areas where the manuscript could benefit from further clarification and elaboration.

Recruitment strategy: It would be helpful to provide more information on the targeting criteria used for the Facebook and Instagram ads. How were the rural zip codes selected? Were there any exclusion criteria, such as language or age restrictions? Additionally, it would be useful to report on the response rate to the ads and the percentage of eligible respondents who agreed to participate.

Sampling strategy: The manuscript could benefit from a more detailed description of the sampling strategy used for participant recruitment. Were participants randomly selected from the eligible pool, or were they purposefully selected based on certain characteristics (e.g., age, income, education)? Providing more information on the sampling strategy would increase the study's transparency and allow for better evaluation of the generalizability of the findings.

Ad design: While the manuscript provides a detailed description of the ad design process, it is not clear how the ad stimuli were selected. Were the 19 ads pre-tested before being presented to participants, or were they selected based on the marketing principles described? Additionally, it would be helpful to provide more information on how the ad stimuli were presented to participants (e.g., in what order, with what instructions).

Qualitative analysis: The manuscript states that two different coders blind-coded each interview, but it is unclear whether they coded the same transcripts or separate transcripts. Additionally, the manuscript could benefit from a more detailed description of the coding process, including the criteria used for developing the codebook and the inter-rater reliability of the coding.

Quantitative analysis: The manuscript reports using linear mixed models to analyze the effect of messenger type and content type on ad ratings, but it is not clear how the models were specified (e.g., which variables were included as fixed and random effects, what covariates were included). Providing more information on the statistical analysis would allow for better evaluation of the study's findings.

Results:

The study captures data from 30 participants and evaluates their reactions to different types of messengers and content featured in the ads. The manuscript presents some interesting findings, but there are several concerns that should be addressed:

Sample size: The sample size is relatively small, with only 30 participants. While this is understandable for a qualitative study, it may not be sufficient for drawing robust conclusions from quantitative analyses.

Generalizability: The study only captures data from a specific group of participants who may not be representative of the broader population. Most participants were white, married, employed, and held a bachelor's degree. This limits the generalizability of the study's findings.

Ads creation: The study does not provide a clear description of how the 19 sample social media ads were created. This raises concerns about the validity of the ads and the extent to which they represent actual ads used in social media.

Messengers and content types: The study's finding that peer messengers and negative outcomes-based content were the most favorably ranked is not surprising, and these findings have been reported in previous studies. The study's contribution is limited as it does not provide a clear explanation of why certain messengers and content types are preferred over others. Additionally, the sample size for each messenger and content type is not provided, making it difficult to draw meaningful conclusions.

Qualitative reactions: While the qualitative reactions to the ads provide some useful insights, the analysis is limited to quotes from a small number of participants. A more comprehensive analysis of the qualitative data, including themes and patterns, would provide a more complete understanding of participants' reactions to the ads.

Bias: The study does not discuss potential biases that may have influenced participants' responses, such as social desirability bias or selection bias.

Overall, while the manuscript presents some interesting findings, the concerns listed above should be addressed before the study can be considered for publication.

Discussion:

The study's limitations are not fully described. The manuscript mentions that there are limitations to the study, but it does not specify what they are. Provide a more detailed description of the limitations, especially since the population studied is highly specific and not representative of the general population.

The manuscript would benefit from a more nuanced discussion of the results. The manuscript provides a summary of the study findings, but it does not delve into the nuances of the results. For example, the manuscript states that respondents were more likely to trust care providers who listened to their needs and affirmed their autonomy, but it does not explain why this is the case. Provide a more detailed discussion of the results, including possible explanations for the findings.

The manuscript could benefit from a more detailed discussion of the implications of the study's findings. While the manuscript briefly mentions the practical implications of the study's findings, it could benefit from a more detailed discussion of how the findings could be applied in clinical care and public health campaigns. Provide a more detailed discussion of the implications of the study's findings, including specific recommendations for care providers and public health officials.

The manuscript should provide more context for the study's contribution to the literature. While the manuscript mentions that the study's findings are in line with other research on vaccine hesitancy, it does not provide enough context to fully understand the contribution of the study to the literature. Provide a more detailed discussion of how the study's findings contribute to the existing literature on vaccine hesitancy among pregnant individuals.

Author Response

We thank Reviewer 1 for your efforts in reviewing the manuscript. We have responded point-by-point to questions and concerns in the attached PDF. 

Reviewer 2 Report

1. Expand on limitations section as there are many (from sample type to study content).

2. Discuss reliability and validity assessment procedures, types, limitations.

3. Key components of the study design utilized are missing (e.g. avoiding of bias strategies, fidelit, etc.

4. What are the implications for practice, research, public health? needs more discussion.

5. Key literature is missing both in introduction and discussion- see latest published studies.

minor editing needed for clarity

Author Response

We thank Reviewer 2 for your efforts in reviewing the manuscript. We have responded point-by-point to questions and concerns in the attached PDF. 

Round 2

Reviewer 1 Report

The manuscript is now in the better form for publication.

Reviewer 2 Report

thanks for the revisions

Minor edits needed